Brain transcriptome sequencing and assembly of three songbird model systems for the study of social behavior

Balakrishnan Christopher N. 1 balakrishnanc@ecu.edu
Mukai Motoko 2 3
Gonser Rusty A. 4
Wingfield John C. 3
London Sarah E. 5
Tuttle Elaina M. 4
Clayton David F. 6
1 Department of Biology and Center for Biodiversity, East Carolina University , Greenville, NC , USA
2 Department of Food Science, College of Agriculture and Life Sciences, Cornell University , Ithaca, NY , USA
3 Department of Neurobiology, Physiology and Behavior, University of California , Davis, CA , USA
4 Department of Biology and The Center for Genomic Advocacy (TCGA), Indiana State University , Terre Haute, IN , USA
5 Department of Psychology, University of Chicago , Chicago, IL , USA
6 Division of Biological & Experimental Psychology, School of Biological and Chemical Sciences, Queen Mary University of London , London , UK
Wink Michael
Electronic publication date: 2014 May 22
Publication date: 2014
Volume: 2
Electronic Location ID: e396
Received 2014 Mar 26; Accepted 2014 May 6
Copyright: © 2014 Balakrishnan et al.
Copyright year: 2014
Copyright holder: Balakrishnan et al.
License: This is an open access article distributed under the terms of the Creative Commons Attribution License, which permits unrestricted use, distribution, reproduction and adaptation in any medium and for any purpose provided that it is properly attributed. For attribution, the original author(s), title, publication source (PeerJ) and either DOI or URL of the article must be cited.
License URL: https://creativecommons.org/licenses/by/4.0/

Keywords: Song learning, Illumina, RNA-seq, Zonotrichia, Song sparrow, White-throated sparrow, White-crowned sparrow, Zebra finch

Funding: NIH 1RC1GM091556 NSF IOS-1010429 University Research Council at Indiana State University NIH 1R01GM084229 East Carolina University Funding for this work was provided by the US National Institutes of Health, National Institute for General Medical Sciences 1RC1GM091556 (to David F. Clayton), the US National Science Foundation, Division of Integrative Organismal Systems 1010429 (to John C. Wingfield), the University Research Council at Indiana State University, NIH 1R01GM084229 (to EM Tuttle and RA Gonser) and East Carolina University. The funders had no role in study design, data collection and analysis, decision to publish, or preparation of the manuscript.

==============================
Emberizid sparrows (emberizidae) have played a prominent role in the study of avian vocal communication and social behavior. We present here brain transcriptomes for three emberizid model systems, song sparrow Melospiza melodia, white-throated sparrow Zonotrichia albicollis, and Gambel’s white-crowned sparrow Zonotrichia leucophrys gambelii. Each of the assemblies covered fully or in part, over 89% of the previously annotated protein coding genes in the zebra finch Taeniopygia guttata, with 16,846, 15,805, and 16,646 unique BLAST hits in song, white-throated and white-crowned sparrows, respectively. As in previous studies, we find tissue of origin (auditory forebrain versus hypothalamus and whole brain) as an important determinant of overall expression profile. We also demonstrate the successful isolation of RNA and RNA-sequencing from post-mortem samples from building strikes and suggest that such an approach could be useful when traditional sampling opportunities are limited. These transcriptomes will be an important resource for the study of social behavior in birds and for data driven annotation of forthcoming whole genome sequences for these and other bird species.

Introduction

The comparative method, broadly speaking, is a powerful approach for understanding adaptations including behavior and central control of physiological responses to environmental change. Natural variation in behavior among species has been used in various taxonomic groups to begin to unravel the molecular underpinnings of animal social behavior. Among these comparative studies of behavior, different strategies and technologies have been deployed in order to gain an understanding of the proximate mechanisms at play. For example, experimental hormonal manipulations and gene sequence comparisons in different species of Microtus voles led to insights into the mechanisms of parental care (Young et al., 1999). Similarly, quantitative trait locus (QTL) mapping studies have recently revealed the genetic architecture of burrowing behavior in Peromyscus mice (Weber, Peterson & Hoekstra, 2013). Phylogenetic analyses of rates of molecular evolution based on transcriptomes in eusocial and solitary bees has also led to insights into potential underpinnings of social behavior variation (Woodard et al., 2011).

Songbirds, or oscine passerines, comprise roughly half of avian diversity and also serve as important models for the study of social behavior. Arguably the most prominent of the songbird species for behavioral research is the zebra finch Taeniopygia guttata, which now boasts a full suite of genomic and molecular tools including a complete genome sequence (Warren et al., 2010), RNA-seq based mRNA (Warren et al., 2010; Balakrishnan et al., 2012) and microRNA data (Gunaratne et al., 2011; Luo et al., 2012), transgenics (Agate et al., 2009) and cell lines (Itoh & Arnold, 2011; Balakrishnan et al., 2012). A key strength of songbirds as a model system, however, has always been the behavioral complexity and diversity of songbirds as a group (Beecher & Brenowitz, 2005; Brenowitz & Beecher, 2005; Clayton, Balakrishnan & London, 2009).

Among songbirds, many comparative neurobiological studies have focused on three species of new world sparrows (emberizidae). Before the zebra finch assumed its role as a model system for vocal learning, Peter Marler and colleagues had demonstrated age-limited song learning and cultural transmission of song dialects in the white-crowned sparrow, Zonotrichia leucophrys (Marler & Tamura, 1964). There is also a striking behavioral polymorphism in which some subspecies, such as Gambel’s white-crowned sparrow Z. l. gambelii, are migratory, living in large non-territorial flocks during non-breeding seasons, whereas other subspecies are non-migratory and are territorial throughout the year (DeWolfe, Baptista & Petrinovich, 1989). White-throated sparrows Zonotrichia albicollis also show polymorphism in behavior but in this case, the polymorphism is known to be caused by a large chromosomal rearrangement on chromosome 2 (Thorneycroft, 1966; Thorneycroft, 1975). Tan morph individuals are homozygotic for the metacentric form of the chromosome whereas white morphs are almost always heterozygous. In addition to coloration, the two morphs differ in a suite of behaviors including increased aggression and promiscuity and decreased parental care in birds of the white morph (Knapton & Falls, 1983; Collins & Houtman, 1999; Tuttle, 2003). Male song sparrows Melospiza melodia are distinctive in that they are territorial during both the breeding season (summer) and much of the non- breeding season (autumn and winter) (Wingfield & Hahn, 1994; Mukai et al., 2009). Different hormonal mechanisms, however, appear to underlie this similar behavioral phenotype with increased plasma testosterone levels driving intensity and persistence of aggression during breeding, but not at other times of year (Wingfield, 1994; Wingfield & Soma, 2002). With this comparative perspective in mind, we have generated brain transcriptomes for these three historically important emberizid songbird models for the study of social behavior: white-throated sparrow, Gambel’s white-crowned sparrow, and song sparrow.

Methods

Sample collection

Samples for each of the three species were collected for diverse research purposes of the laboratories involved, so sampling strategy for each species was unique. Animal procedures were approved by the Institutional Animal Care and Use Committees of the University of California, Davis (protocol 07-13208) and the University of Illinois (protocol 11062) and were conducted in accordance with the NIH Guide for the Principles of Animal Care.

White-throated sparrow

During migration, white-throated sparrows and other birds are often killed in collisions with buildings. We took advantage of this unfortunate fact by sampling white-throated sparrows that had been opportunistically collected following night migration and collision into McCormick Place, Chicago, IL. Birds that had been killed overnight were collected first thing in the morning beginning at dawn by David Willard, Collection Manager—Birds, Field Museum of Natural History, Chicago, IL. Specimens used in this study were collected during the spring migration in 2010. Each specimen was immediately vouchered at the Field Museum where they were dissected to determine sex. Whole brain tissue was stored in RNA-later (Life Technologies, Carlsbad, CA). Prior to analysis we determined the morph of each bird sampled using a modification of Michopoulos et al. (2007), which is based on the identification of a morph-specific SNP present in the vasoactive intestinal peptide (VIP) gene. We modified the protocol by using labelled PCR primers, so that the amplification products could be analyzed on an ABI PRISM Genetic Analyzer (Life Technologies). For RNA sequencing we used the brains from six males, three white and three tan.

Gambel’s white-crowned sparrow

We captured Gambel’s white-crowned sparrows within the University of California, Davis campus in February 2008, using Potter traps baited with seed, and determined their sex using published PCR methods (Griffiths et al., 1998). After two weeks of acclimation in captivity we anesthetized 12 male birds with with isoflurane, decapitated them and collected the whole hypothalamus from each bird. After dissection we immediately froze the samples in liquid nitrogen. Fieldwork in California was covered by the US Fish and Wildlife permit (MB713321-0) and State of California permit (SC-004400).

Song sparrow

Between July and August 2011 we captured seven male song sparrows using song playbacks from behind a mist net. We conducted fieldwork at two locations in central Illinois: “Phillips Tract” (40 07′ 54.74″N 88 08′ 39.66″W) and Vermillion River Observatory (40 03′ 50.79″N 87 33′ 30.30″W). We euthanized the birds immediately following capture in the net, and then dissected auditory forebrain tissue (auditory lobule, or AL). AL is a composite brain area including the caudomedial nidopallium (NCM), caudomedial mesopallium (CMM) and Field L and can be readily dissected following bisection of the brain along the midline (Cheng & Clayton, 2004). We immediately froze the specimens on dry ice. Flat skins of collected song sparrows have been accessioned in the Illinois Natural History Survey, Urbana, Illinois. We conducted fieldwork in Illinois under US Fish and Wildlife Service Permit SCCL-41077A.

RNA extraction, library preparation and sequencing

White-throated sparrow and song sparrow

In order to broadly describe the brain-expressed transcriptome of the white-throated sparrow, we extracted RNA from whole brain. We homogenized the entire brain in Tri-Reagent (Molecular Research Center, Cincinnati, OH) for RNA purification and extracted total RNA following the Tri-Reagent protocol. We then DNase treated (Qiagen, Valencia, CA) the total RNA to remove any genomic DNA contamination, and further purified the resulting RNA using Qiagen RNeasy columns. We assessed the purified total RNA for quality using an Agilent Bioanalyzer (Agilent Technologies, Wilmington, DE). Library preparation and sequencing were done at the University of Illinois Roy J. Carver Biotechnology Center. The RNAseq libraries were constructed with the Illumina TruSeq RNA Sample Prep Kit (Illumina, San Diego, CA). Briefly, polyA+ messenger RNA was selected from 1ug of total RNA and chemically fragmented. First-strand cDNA was synthesized with a random hexamer and SuperScript II (ThermoFisher, Waltham, MA). After second-strand synthesis, the double-stranded DNA was blunt-ended, 3′-end A tailed, ligated to barcoded adaptors and amplified with 15 cycles of PCR using Kapa HiFi polymerase (Kapa Biosystems, Woburn, MA). The six barcoded libraries were quantitated with Qubit (ThermoFisher) and the average size was determined on a Bioanalyzer DNA7500 DNA chip (Agilent). The libraries were pooled in equimolar concentration and the pool was quantitated by qPCR on an ABI 7900HT (ThermoFisher). Sequencing was done in a single lane of an Illumina HiSeq2000 using a TruSeq SBS sequencing kit version 3. Fastq files were demultiplexed and generated with the software Casava 1.8.2 (Illumina). The same basic procedure was used to sequence the song sparrow except for the fact that we extracted RNA from the dissected AL (rather than whole brain) tissue, and that samples from seven individuals were run in a single lane of paired end (rather than single end) sequencing.

Gambel’s white-crowned sparrow

We extracted total RNA from each hypothalamus using TRIzol reagent (Life Technologies) followed by RNA cleanup using Qiagen RNeasy Mini Kits. We then pooled RNA samples, quantified them using a Nanodrop (ThermoFisher) and ran them on a Bioanalyzer for quality control (RIN = 8.5). We used this pooled RNA sample to generate a mRNA-seq library of 400 bp size with a mRNA-Seq 8 sample prep kit (Illumina) following manufacturer’s protocol with slight modifications. We began by isolating mRNA using oligo(dT) and then fragmented it using divalent cations under elevated temperature. We then reverse transcribed the RNA into cDNA using random primers, modified and ligated with GEN PE adapters. We ran the resulting cDNA on an agarose gel, excised a 400 bp band and enriched the library with 15 cycles of PCR. We validated the final library using a Bioanalyzer and confirmed a distinct band at approximately 400 bp. Pair-end sequencing (100 bp × 2) was performed by the Genome Center DNA Technologies Core at the University of California, Davis, using an Illumina HiSeq 2000 and TruSeq SBS kit version 2.

Zebra finch

To provide a benchmark for comparison, we compared our newly collected data with previously published data from zebra finches Taeniopygia guttata (Balakrishnan et al., 2012, GenBank Accession: SRX493920–SRX493922). These data were derived from RNA extracted from the AL of female zebra finches. The three libraries were derived from pools of 10 female finches each, and sequenced on an Illumina Genome Analyser and processed with Illumina pipeline 1.6.

Transcriptome assembly, annotation and assessment

We checked overall sequence quality using FastQC (http://www.bioinformatics.babraham.ac.uk/projects/fastqc/) and trimmed reads using ConDeTriV2.2 (Smeds & Kunstner, 2011). We used default settings for trimming except for the high quality (hq) threshold which was set to 20 and lfrac, the maximum fraction of reads with quality <10, which was set to 0.2. The lfrac parameter allows for trimming, rather than complete removal, of reads with low quality ends.

We used the Trinity (version r20131110) assembler (Grabherr et al., 2011) to generate de novo assemblies for each species. For white-throated sparrow we assembled the reads for the two color morphs both separately and combined. Assembling the reads separately was reasonable given evidence of sequence divergence within the inversion (Thomas et al., 2008) and assembling the reads together was reasonable to improve coverage outside such areas. We used default settings in Trinity besides those specific to our computing system (we generally used 24 CPUs and allowed for 100G of memory). We used TransDecoder (included in the Trinity package) to identify open reading frames (ORFs) in our predicted transcripts.

We assessed the quality of our assembly by estimating N50 and average transcript length. The shortcomings of such metrics for transcriptome assessment have been described (O’Neil & Emrich, 2013) and we use them here primarily to facilitate comparison with previously published studies. To provide further insight into assembly quality, we also assessed 5′ to 3′ gene model coverage relative to annotated zebra finch genes (see details below) and quantified the number of transcripts containing both start and stop codons using the annotation information provided by TransDecoder (“type:complete” in the fastq header).

We used BLAST (Altschul et al., 1990) searches against a database of Ensembl (release 74) zebra finch transcripts to annotate our ORF-containing transcripts. Functional description of annotated transcripts was conducted using Gene Ontology, and statistical over and under representation was tested using CORNA software (Wu & Watson, 2009) and Fisher’s Exact Tests with p values adjusted for multiple testing (Benjamini & Hochberg, 1995). For each assembly we tested our identified set of putative zebra finch orthologs relative to the full population of Ensembl transcripts.

Gene expression and read-mapping profiling

In order to compare read mapping and gene expression profiles across libraries, we mapped RNA-seq reads to the zebra finch whole genome assembly (2.3.4) using Stampy, a read mapper tailored for divergent reads relative to the reference genome (Lunter & Goodson, 2011). We mapped reads for all six individual white-throated sparrows, three of the seven song sparrows, and the pooled white-crowned sparrow using default settings but with the substitution rate set to 0.05 to accommodate sequence divergence. In addition, we mapped reads from previously published zebra finch auditory forebrain reads (Balakrishnan et al., 2012) using substitution rate = 0.01.

To quantify gene expression, we used htseq-count (Anders, Pyl & Huber, 2014) to tally reads relative to Ensembl gene models and then normalized the read counts using the regularized log transformation in DE-Seq2 (Anders & Huber, 2010). Expression profiles were then visualized by Euclidean distance based clustering and principal components analysis (PCA) using heatmap.2 in the gplots R package, and the plotPCA function in DE-Seq2. We then also used the geneBody.py script within the RseqC package (Wang, Wang & Li, 2012) to describe read coverage across gene models and to test specifically for a 3′ bias in transcript coverage in post-mortem samples.

Results & Discussion

RNA extraction and sequencing

Despite collecting tissues for the white-throated sparrow opportunistically from building strikes, we were able to extract reasonably high quality RNA from all samples (Fig. 1). This finding suggests that post-mortem collected birds can be used as a viable source of RNA for transcriptome sequencing. From a total of twelve samples, we selected a set of six (three per morph) with Bioanalyzer RNA integrity numbers (RIN) above 7 (10-083 (7.2), 10-092 (7.2), 10-093 (7.7) and 10-118 (8.5), 10-124 (8.0) and 10-308 (7.9)). Samples for sequencing were also chosen such that tan and white morphs were collected at the same time of year (spring migration 2010). By chance, our tan samples had higher average RINs than the white morph samples did (Fig. 1). RNA from the other two species were of good quality and met Illumina’s standard QC benchmark of RIN > 8. All of our sequencing runs yielded high quality sequence data. After fairly stringent quality trimming, we retained over 89% of the initial nucleotides sequenced (Table 1). Raw RNA seq reads have been deposited to the GenBank Short Read Archive under accession numbers SRX342288–SRX342293, SRX493875–SRX493882, and SRX514971.

Table 1 RNA-seq dataset.

Raw number of reads and bases before and after trimming with ConDeTri.

Species	Reads before	Bases before	Paired reads after	Paired read bases after	Single reads after	Single read bases after	
WTSP-Tan	99,374,744	9,937,474,400	NA	NA	97,162,587	9,014,814,467	
WTSP-White	97,605,312	9,760,531,200	NA	NA	95,347,015	8,779,352,471	
SOSP-Paired	271,249,550	27,124,855,000	245,289,038	23,613,455,033	11,228,223	992,474,010	
WCSP-Paired	160,229,712	16,022,971,200	153,636,836	14,171,465,431	2,871,235	213,815,184	

Figure 1 RNA quality from post-mortem sampled sparrows.

Bioanalyzer gel image showing RNA extracted from 12 white-throated sparrows sampled post-mortem. RNA integrity numbers (RIN) are given at the bottom and ranged from 6.4 to 8.5. Samples chosen for sequencing are indicated by tan and white circles, representing tan and white morph sparrows, respectively.

Transcriptome assembly and annotation

We reconstructed a large number of transcripts (>95,000) and open reading frame (ORF) containing transcripts (>54,000) in all of our assemblies, exceeding the likely number of coding genes (Table 2). These transcripts reflect a combination of partial transcripts, alternative isoforms, allelic variants, and noncoding transcripts. We were able to generate high quality transcriptomes based on N50 and average transcript length (Table 2). N50s for the assemblies were 1,942 for the white morphed white-throated sparrow, 2,557 for the tan morph, 3,415 for Gambel’s white-crowned sparrow and 4,072 for the song sparrow (Table 2). For the song sparrow, this is an improvement over a recent 454-based transcriptome (N50 = 482; Srivastava et al., 2012). As expected, N50 in general improved with increased sequencing depth (with paired end data sets benefitting from both the reads being paired and having more reads). One exception to this rule was in the white-throated sparrow, where combining reads from the two morphs actually generated a worse assembly in terms of N50 relative to the “tan morph only” assembly (combined N50 = 2,284, tan only N50 = 2,557). Tan morph individuals are homozygous for a large structural polymorphism spanning much of chromosome 2 whereas white morph individuals are heterozygous. Recombination within the inversion is suppressed, allowing genetic divergence in this region (Thomas et al., 2008), and potentially explaining the drop in N50 in the combined assembly. For the purposes of annotation of the white-throated sparrow we therefore used the two morph-specific assemblies, merging them after the assembly process.

Table 2 Transcriptome assembly description.

Tissue of origin, pool size, assembly statistics (N50, average transcript length, number of transcripts) and annotation description (number of zebra finch genes with significant BLAST hit) for whole assembly and open reading frame (ORF) containing transcripts. “Complete Transcripts” are those containing both a start and stop codon. We used the individual tan and white morph assemblies in the subsequent BLAST search and annotation which yielded 15,805 genes.

Species	Tissue	Pool size	N50	Mean length	# Transcripts	# ORF	Complete transcripts	ZF genes	
WTSP-Tan	Whole brain	3	2,557	1,119	116,894	54,868	22,799	–	
WTSP-White	Whole brain	3	1,942	960	95,129	37,910	11,855	–	
WTSP-Both	Whole brain	6	2,284	982	149,184	58,284	24,388	15,805	
SOSP	Auditory forebrain	7	4,072	1,416	276,670	133,740	79,451	16,864	
WCSP	Hypothalamus	12	3,415	1,591	307,617	206,926	115,515	16,646	

Although N50s were generally high, the white-throated sparrow assemblies, which were based on smaller, single-end datasets and post-mortem samples, had the lowest scores. This effect was even more dramatic when assemblies were assessed in terms of the number of complete transcripts possessing both a start and stop codon. Gambel’s white-crowned, song, and white-throated sparrow transcriptomes contained 115,515, 79,451, and 24,388 complete transcripts, respectively (Table 2). Because the white-throated sparrow samples were collected post-mortem and had the fewest reads, we cannot determine whether post-mortem sampling itself influenced assembly quality. Given the relatively high quality (RINs) of the white-throated sparrow RNA, however, it is more likely that the reduced quality of the assembly is a result of it being generated from a smaller dataset.

For white-throated sparrow we were able to find predicted transcripts with significant BLAST hits to 15,805 zebra finch genes (89% of Ensembl annotated zebra finch genes), whereas for song sparrow we found 16,846 (94%) and Gambel’s white-crowned sparrow 16,646 (93%). Therefore, in terms of unique BLAST hits, the song sparrow and Gambel’s white-crowned assemblies were also better than that of the white-throated sparrows. All three assemblies, however, cover a large proportion of known genes and represent an improvement of over recent 454-based bird transcriptomes (e.g., violet-eared waxbill, 11,084 genes, Balakrishnan et al., 2013).

We evaluated and compared the general composition of genes present in each of the new assemblies by performing a Gene Ontology (GO) analysis, using the GO annotation of the complete zebra finch genome as the point of reference for the statistical tests of enrichment (Table 3). All three datasets shared a number of similarities, including significant enrichment for eight GO categories (“cytoplasm”, “intracellular’, “mitochondrion”, “nucleic acid binding”, “nucleolus”, “protein binding”, “protein phosphorylation” and “transferase activity”) and under-representation of six categories (“cytokine activity”, “DNA integration”, “extracellular region”, “hormone activity”, “immune response” and MCH Class I protein complex”). The under-represented categories may in part reflect the well-described pattern of limited immune activity, or “immune privilege” in the brain (Galea, Bechmann & Perry, 2007). As in previous studies of avian brain gene expression, however, we did see some evidence of expression of the MHC Class I gene itself (Ekblom et al., 2010; Balakrishnan et al., 2013).

Table 3 Functional description of transcriptome assemblies.

Gene Ontology categories significantly (A) over- and (B) under-represented in song (SOSP), white-crowned (WCSP) and white-throated (WTSP) sparrows (observed/expected, FDR adjusted Fisher’s exact test, p < 0.05).

GO Category	SOSP	WCSP	WTSP	
A	
cytoplasm	1810/1739	1793/1718	1751/1650	
intracellular	1629/1575	1632/1555	1577/1494	
mitochondrion	790/753	788/744	781/715	
nucleic acid binding	935/903	935/892	900/857	
nucleolus	244/231	243/229	241/220	
protein binding	5298/5218	5258/5154	5037/4951	
protein phosphorylation	558/539	558/532	542/511	
transferase activity, transferring
phosphorous containing groups	538/519	538/513	522/493	
B	
cytokine activity	43/58	40/58	37/55	
DNA integration	8/13	7/13	4/12	
extracellular region	263/320	264/316	238/303	
hormone activity	31/43	32/43	26/41	
immune response	68/88	61/87	57/84	
MHC Class I protein complex	3/8	2/7	2/7	

Interestingly, genes annotated with the GO term “olfactory receptor activity” are well represented in all three assemblies (where observed/expected were 165/150 in white-throated sparrows, 165/156 in song sparrow, and 165/158 in Gambel’s white-crowned sparrow, out of a total of 168 annotated genes). This was notable as a previous 454-based whole brain transcriptome of another songbird did not detect any olfactory receptor genes at all (Balakrishnan et al., 2013). The detection of such genes here suggests that the increased sequencing depth provided by the Illumina platform has aided in this regard. Despite the generally tissue-restricted distribution of olfactory receptor expression, we were able to pick up these genes in all of our tissue samples irrespective of the brain region targeted. High depth RNA-sequencing data including those presented here will therefore be useful for annotating these diverse olfactory receptor transcripts.

Thirteen other GO terms were significantly under-represented only in the white-throated sparrow assembly (Table 4). These categories were relatively well-represented in the other two sparrow assemblies (Table 4) and included “visual function”, “G-protein coupled receptor activity”, and “neurotransmitter transport”. The white-throated sparrow assembly differs from the others in several factors that could contribute to this difference in gene composition, including tissue of origin (whole brain, versus auditory forebrain or hypothalamus), physiological condition (spring migration, versus breeding season or captive/wintering) and post-mortem tissue collection.

Table 4 Functional differences between post-mortem and fresh tissues.

GO terms underrepresented in post-mortem white-throated sparrow samples (observed/expected, adjusted p < 0.01), but not in song sparrow and white-crowned sparrow (adjusted p > 0.05).

GO category	WTSP	WCSP	SOSP	
photoreceptor activity	3/12	10/13	9/13	
protein-chromophore linkage	3/12	10/13	9/13	
visual perception	7/18	16/19	15/19	
response to stimulus	7/17	14/18	13/18	
G-protein coupled receptor activity	345/381	391/397	389/402	
G-protein coupled purinergic
nucleotide receptor activity	11/21	18/22	18/23	
G-protein coupled purinergic
nucleotide receptor signaling pathway	11/21	18/22	18/23	
transporter activity	136/157	153/164	157/166	
receptor activity	497/532	552/554	551/561	
G-protein coupled receptor signaling pathway	463/496	513/517	514/523	
integral to membrane	1564/1617	1683/1687	1692/1704	
neurotransmitter transport	16/24	23/25	21/25	

Transcriptome coverage of zebra finch gene models

We performed further analysis of read distribution and the relative abundance of different transcripts in each of the source tissues, by mapping RNAseq reads back to the zebra finch genome reference. For comparison we also included previously published RNAseq read data from the zebra finch auditory forebrain (Balakrishnan et al., 2013). White-throated sparrow reads mapped at a lower rate (average = 83% of reads mapped) than reads from Gambel’s white-crowned sparrow (90%), song sparrow (94%) and zebra finch (93%). Among the reads that did map to the genome, however, all of the species were similar in showing a large proportion of reads (53.2 +/− 3.6%) mapping outside of currently defined zebra finch genes, suggesting extensive transcription outside of known genes.

Based on this read mapping we were able to assess coverage of annotated genes. This was important given our post-mortem sampling of white-throated sparrows. In highly degraded samples we would expect to see a strong 3′ bias in gene coverage. RNA quality as measured by RIN was only slightly lower in white-throated sparrow samples and thus, we found that 3′ bias was similar across all of our samples (Fig. 2). This finding further suggests that RNA degradation may not be the primary factor associated with the lower assembly quality in the white-throated sparrow assembly.

Figure 2 Coverage of zebra finch gene models by RNA-seq reads.

Gene model coverage across all genes based on mapping of reads to the zebra finch genome. Samples collected post-mortem from white-throated sparrow show a similar gene coverage profile to freshly collected samples. Zebra finch data included fewer total reads, explaining the lower depth across genes.

Cheviron, Carling & Brumfield (2011) documented the time course of RNA degradation post-mortem, and also suggest that such samples can provide a useful source of RNA, even though such specimens are often overlooked. Similarly, a recent RNA-sequencing study of pinnipeds successfully used post-mortem samples (Hoffman et al., 2013). Although clearly not an ideal strategy for studies aimed at quantifying gene expression, the use of recently killed samples is viable strategy for initial transcriptome description, and in our study gave access to a large portion of the transcriptome. This approach could be particularly useful for rare species where collection of fresh specimens is impossible.

Impacts of ancestry, tissue of origin, and library preparation on expression profile

We used clustering analysis to compare the broad structure of gene expression in the different samples, recognizing that the samples differed in multiple dimensions (i.e., species, sex, brain region, physiological condition, collection method, sequencing method). If species or sex were the dominant factors driving the differences in gene expression patterns, one would expect to see a clustering pattern with zebra finch as the most divergent profile (Fig. 3A). Similarly, if the sequencing facility and platform were dominant technical factors one would expect to see either the zebra finch or the white crowned-sparrow as most divergent (Fig. 3B). However, the zebra finch samples clustered closely with the song sparrow samples taken from the same brain region (auditory forebrain), with the white throated sparrow samples from the whole brain clustering together as most divergent, and the Gambel’s white-crowned sparrow samples from hypothalamus in between (Figs. 3C and Fig. 4). This echoes previous findings that brain region is a major determinant of gene expression pattern in songbirds (Replogle et al., 2008; Drnevich et al., 2012). Both euclidean distance-based clustering and PCA also highlight the fact that zebra finches, which were sacrificed in captivity and sequenced in pools of ten, had much reduced variance in expression profile relative to our non-pooled, field-collected white-throated sparrow and song sparrow samples (Fig. 4).

Figure 3 Alternative expectations for expression profile clustering.

Alternative expectations if (A) phylogeny or sex (B) sequencing platform or library preparation protocols or (C) tissue of origin, were the dominant factor underlying expression clustering. Only tissue of origin unites zebra finch and song sparrow samples together as observed in the clustering analysis (Fig. 4).

Figure 4 Clustering of expression profiles from four songbird species.

(A) Hierarchical clustering and (B) Principal components analysis of expression profiles for six white-throated sparrow (WTSP), three song sparrow (SOSP), three zebra finch (ZF) and one white-crowned sparrow libraries. Libraries derived from auditory lobule (AL) tissue cluster (SOSP and ZF) to the exclusion of the others. White-throated sparrow samples, taken from whole brain (rather than forebrain as the other samples are) show divergent and variable profiles. Zebra Finch (ZF) samples collected in captivity and generated from pools of 10 individuals, show much reduced sample variability.

Conclusion

Transcriptome assemblies are a valuable resource, particularly for species without reference genomes, providing access to a large proportion of the coding and noncoding expressed genome. For taxa with genomes, or with genomes in progress, transcriptome data provides empirical (as opposed to model based) information on transcript structures including alternative isoforms that are not well-annotated in most species. We have presented here neuro-transcriptomic data for three important model species for the study of social behavior and neurobiology building on a growing body of such data (e.g., Balakrishnan et al., 2013; Ekblom et al., 2014; MacManes & Lacey, 2012; Moghadam et al., 2013).

Supplemental Information

Supplemental Information 1 Song sparrow gene ontology

Statistical tests of Gene Ontology (GO) category over and under representation in the song sparrow transcriptome assembly.

Click here for additional data file.

Supplemental Information 2 White-throated sparrow gene ontology

Statistical tests of Gene Ontology (GO) category over and under representation in the white-thrated sparrow transcriptome assembly.

Click here for additional data file.

Supplemental Information 3 Gambel’s white-crowned sparrow gene ontology

Statistical tests of Gene Ontology (GO) category over and under representation in the Gambel’s white-crowned sparrow transcriptome assembly.

Click here for additional data file.

Thanks to Matt MacManes for helpful feedback on the preprint version of this paper. David Willard (Collection Manager—Birds, Field Museum of Natural History, Chicago, IL) collected and provided access to white-throated sparrow tissues used in this study. Antonio Celis Murillo provided invaluable assistance with fieldwork on song sparrows in Illinois.

Additional Information and Declarations

Competing Interests

Author Contributions

Animal Ethics

Field Study Permissions

DNA Deposition

The authors declare there are no competing interests.

Christopher N. Balakrishnan conceived and designed the experiments, performed the experiments, analyzed the data, contributed reagents/materials/analysis tools, wrote the paper, prepared figures and/or tables, reviewed drafts of the paper.

Motoko Mukai, Sarah E. London and Elaina M. Tuttle conceived and designed the experiments, performed the experiments, contributed reagents/materials/analysis tools, wrote the paper, reviewed drafts of the paper.

Rusty A. Gonser conceived and designed the experiments, contributed reagents/materials/analysis tools, reviewed drafts of the paper.

John C. Wingfield and David F. Clayton conceived and designed the experiments, contributed reagents/materials/analysis tools, wrote the paper, reviewed drafts of the paper.

The following information was supplied relating to ethical approvals (i.e., approving body and any reference numbers):

All animal procedures were approved by the Institutional Animal Care and Use Committees of the University of California, Davis (07-13208) and the University of Illinois (11062) and were conducted in accordance with the NIH Guide for the Principles of Animal Care.

The following information was supplied relating to field study approvals (i.e., approving body and any reference numbers):

Illinois: US Fish and Wildlife Service (SCCL-41077A); California: US Fish and Wildlife Service (MB713321-0); and State of California (SC-004400).

The following information was supplied regarding the deposition of DNA sequences:

GenBank: SRX342288–SRX342293, SRX493875–SRX493882, SRX514971.

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
