# Peer review of "Brain transcriptome sequencing and assembly of three songbird model systems for the study of social behavior"

_PeerJ, doi:10.7717/peerj.396_

## Round 0.1 · original submission · Major Revisions

Dear authors

Thank you for submitting your manuscript to our journal. As you see our reviewers suggest a revision of your ms. If you are willing to do so, we would be happy to reconsider your revised manuscript.

Reviewer 1 ·

Basic reporting

Balakrishnan et al. generated transcriptomes for three model songbird systems. The paper is clear, well-written, and provides a new set of the data that will be of general interest. They also build on previous work and show the impact of post-mortem decay on RNA isolation. The transcriptomic variation within the White-throated sparrow is of concern, but the type of variation they observed seems to be typical of transcriptomic studies on wild animals. Thus, I don’t think it can be used as a major criticism of the paper. The study design, analyses, and interpretations are straightforward. Most of comments are minor and are aimed to improve the clarity of the text. Otherwise, the paper, in my opinion, is ready to be published after minor edits.

Experimental design

See comments below

Validity of the findings

No comments

Additional comments

Comments/Suggestions:

- The methods switch between active and passive voice. To be consistent pick one.

- To stay consistent I recommend including the subspecies names or state whether a species is monotypic when the species are introduced in the methods

Line 75 We took advantage of this unfortunate fact by collecting birds

- The authors didn’t collect the birds? Obtaining samples…..
Lines 81-82 Prior to analysis we determined the morph of each sampled bird using a modification of Michopoulous et al. (2007)

- What is the method of Michopoulous et al. (2007) and what were the modifications?

Line 108 - Library preparation was done using Illumina TruSeq RNA Sample Prep Kit and manufacturer’s protocols

- Can a brief summary of that protocol be included? The kits change frequently and methods used in this study will have a longer lifespan if readers can follow what was done after the kit is gone.

Line 120 What manufacture’s protocol? This isn’t referring to the Qiagen RNeasy Mini Kit, correct?

Line 139 - settings in Trinity besides those specific to our computing system (memory allocation, etc.).
- Include the specifics pertaining to your computing system

Line 179 integrity numbers (RIN) above 7 (10-083
- missing or extra (

Methods – Is it worth mentioning any particular bias that you would expect or know about regarding the construction of libraries using the mRNA-seq library kit (Used for White-crowned sparrow) versus Illumina TruSeq RNA Sample Prep Kit?
I am not sure what kit was used for the White-crowned sparrow.

Line 182 Remind the reader why the tan morph samples would have higher RINs

Line 193 – Can you provide N50s for all taxa?

Line 268 biological effects of phylogeny and Line 269 If phylogeny
- You should define what you mean by phylogeny or use general terms. I assume you are referring to shared common ancestry

Line 273 We did not conduct statistical tests of differential gene expression
- Isn’t this going to be an issue on sequencing transcriptomes on wild animals? I am sure one can control for some of the induced variation, but it is has to be really difficult to control.

Reviewer 2 ·

Basic reporting

In this manuscript, the authors report the transcriptome sequences of brain areas of three songbird species that were produced with RNA sequencing. Further, they propose that such transcriptomes can be produced from bird brains collected post mortem. The authors conclude that “postmortem” transcriptomes are of acceptable quality and that the composition of brain transcriptomes are affected brain area and species.

In general this is a worthy paper since (1) the transcriptomes of these species are interesting to the songbird community and (2) since the use of brains of specimens collected post mortem would facilitate the study of rare species and increase the acceptance of neuromolecular works by the general public.

The paper suffers from three larger problems (see below) that could be solved quite easily.

Experimental design

Area and species differences: Since the transcriptomes of the three species originate from different brain parts or from the entire brain the authors include transcriptome data from brain areas of the zebra finch for comparisons. The origin of these data is, however, unclear. Zebra finch transcriptome sequencing is not mentioned in the methods part. In case that these data have been published/used before, statistical corrections are needed. In case that the authors remove the zebra finch data, the conclusion that “tissue of origin is a primary determinant of expression profile” needs to be scaled back throughout.

Postmortem transcriptomes: Unfortunately the authors do not analyze a freshly-frozen brain of the same species (white-throated sparrow) as compared to the brains collected after longer postmortem intervals. In consequence, it is difficult to estimate the quality of postmortem transcriptomes. In relation, it is not possible to interpret Table 4. Since such a control might be difficult to produce with the white-throated sparrow, the authors could easily produce RNAseq data from brains collected from postmortem zebra finches.

Sequencing procedures: The transcriptomes of the different species were produced in part with different techniques. This is not ideal but acceptable. It is, however, unclear how the sequencing was performed: The authors state that they ran single lanes after pooling of the libraries (lines 110/111, line 114/115). However, they do not mention multi-plexing. Thus, it is difficult to understand how they end up with six transcriptomes in case of the white-throated sparrow (depicted in Figure 2) and with three in case of the song sparrow (in the latter they include 7 brain areas, line 114) (depicted in Figure 2).

Minor queries:
How was the auditory lobe defined/dissected?

Validity of the findings

Area and species differences: In case that the authors remove the zebra finch data, the conclusion that “tissue of origin is a primary determinant of expression profile” needs to be scaled back throughout.

Postmortem transcriptomes: Since this essential control (tissue frozen collected immediately after death as compared to tissue collected after longer postmortem intervals) is missing, it is difficult to estimate the quality of postmortem transcriptomes. In relation, it is not possible to interpret Table 4. Since such a control might be difficult to produce with the white-throated sparrow, the authors could easily produce trancriptomes from brains collected from postmortem zebra finches.

Minor queries:
GO-analysis: The authors report a rather short list of GO-terms that are significantly over and under-represented in each of the three species (Table 3). Is this list exhaustive?

I don’t understand how the time of the year (physiological state of the animals) affects the RNA quality of white but not of tan morph samples (lines 181/182)?

Additional comments

The most interesting point of the paper is the use of postmortem brain tissue. I suggest that the authors improve the paper as indicated above.

---

## Round 0.2 · accepted · Accept

Dear authors

Thank you for resubmitting your manuscript to our journal. We are happy to accept your ms by now. Thank you for submitting your research results to our journal.